# Growth and rapid succession of methanotrophs effectively limit methane release during lake overturn

Magdalena J. Mayr [1,2], Matthias Zimmermann [1,2], Jason Dey[1], Andreas Brand[1,2], Bernhard Wehrli[1,2] & Helmut Bürgmann [1✉]

Lakes and reservoirs contribute substantially to atmospheric concentrations of the potent greenhouse gas methane. Lake sediments produce large amounts of methane, which accumulate in the oxygen-depleted bottom waters of stratified lakes. Climate change and eutrophication may increase the number of lakes with methane storage in the future. Whether stored methane escapes to the atmosphere during annual lake overturn is a matter of controversy and depends critically on the response of the methanotroph assemblage. Here we show, by combining 16S rRNA gene and *pmoA* mRNA amplicon sequencing, qPCR, CARD-FISH and potential methane-oxidation rate measurements, that the methanotroph assemblage in a mixing lake underwent both a substantial bloom and ecological succession. As a result, methane oxidation kept pace with the methane supplied from methane-rich bottom water and most methane was oxidized. This aspect of freshwater methanotroph ecology represents an effective mechanism limiting methane transfer from lakes to the atmosphere.

[1] Eawag, Swiss Federal Institute of Aquatic Science and Technology, 6047 Kastanienbaum, Switzerland. [2] Institute of Biogeochemistry and Pollutant Dynamics, Department of Environmental Systems Science, ETH Zurich, 8092 Zurich, Switzerland. ✉email: helmut.buergmann@eawag.ch

Lakes and impoundments emit a greenhouse gas equivalent of 20% of the global fossil fuel $CO_2$ emissions, with methane contributing 75% of these $CO_2$ equivalents[1]. Stratified lakes accumulate the potent greenhouse gas methane in their oxygen-depleted bottom waters[2]. During lake overturn, stored methane may reach the surface layer, thereby running the risk of out-gassing. Despite the established importance of lakes for global greenhouse gas emissions, the fate of methane during the over-turn period is still a matter of controversy. Two competing hypotheses regarding the fate of accumulated methane have been proposed. According to the first line of thinking, most of the stored methane will be released into the atmosphere[3] as an emission pulse that adds to continuous methane fluxes across the water-air interface. According to the second hypothesis, metha-notrophs oxidize most of the methane with oxygen, thereby forming biomass and $CO_2$, a greenhouse gas which has a 34 times lower global warming potential for a 100-yr time-scale[4]. Some previous estimates assumed overturn to occur on very short time scales, e.g. a single day[3] therefore favoring the first hypothesis. Other studies have indicated that lake overturn typically takes place on a scale of weeks to months, even in shallow lakes[5–7], which may allow time for methane oxidation. This controversy has not yet been fully resolved[8] and the role and ecological dynamics of methane-oxidizing bacteria (MOB) during lake overturn remains to be explored.

MOB in lakes have mostly been investigated during the stra-tified season, when a structured MOB assemblage forms an effi-cient methane converter preventing the methane accumulating in the bottom water from outgassing[9–12]. During lake overturn environmental conditions change compared to the stratified situation[13]. How the MOB assemblage responds to lake overturn remains unknown, but its growth rate and resulting methane-oxidation capacity, which are critical for the amount of methane emitted, can be expected to change in this period. The lake cools down and oxygen and methane, which are vertically separated during stratification, become simultaneously available as water with different substrate concentrations mix in the expanding surface layer. Taipale et al. noted an increase of methane-oxidation rate and abundance of type I methanotrophs linked to the autumn lake mixing period[5], but the dynamics of different MOB taxa have not been studied.

Here we present a field study covering the entire three months of the autumn lake overturn of shallow eutrophic Lake Rotsee, Switzerland. We asked firstly if the MOB assemblage grows fast enough to oxidize the methane mobilized from the bottom water before outgassing and secondly whether the standing MOB assemblage is activated, or a new assemblage successively takes over in the changing lake. We used 16S rRNA gene amplicon sequencing, *pmoA* mRNA sequencing and qPCR, CARD-FISH and potential methane-oxidation rate measurements to investi-gate succession, growth and methane-oxidation capacity of the MOB assemblage during lake overturn. The rates of physical mixing and the transfer and transformation of methane have been analyzed with a process-based model in a parallel study[14].

The present study provides detailed insights into the dynamics of freshwater lake MOB during the critical overturn period. We show that a new and highly abundant MOB assemblage, repre-senting up to 28% of 16S rRNA gene sequences, thrived in the expanding mixed layer. In parallel with MOB abundance, the methane-oxidation capacity of the lake increased substantially, thereby limiting methane emissions to a small percentage of the stored methane.

## Results

**Characteristics of stratification and overturn dynamics.** To determine the response of the MOB assemblage and methane-oxidation activity to lake overturn, we sampled the shallow eutrophic Lake Rotsee at eight time points covering stratification, overturn and inverse stratification. The starting point of our sampling campaign on Oct 4, 2016 represents a typical stratified situation, which in Rotsee starts to develop in May[15]. In the following, we refer to the large amounts of methane that have accumulated during summer in the hypolimnion as 'stored methane'.

Most of the oxygen and stored methane in the water column were vertically separated from each other prior to lake overturn (Fig. 1a). The lake water cooled down from October to January, resulting in a gradual expansion of the mixed layer until December (Fig. 1b). By Dec 12, a cooler mixed layer formed on top of warmer bottom water (Fig. 1b), which has a slightly higher salinity (Supplementary Fig. 1).

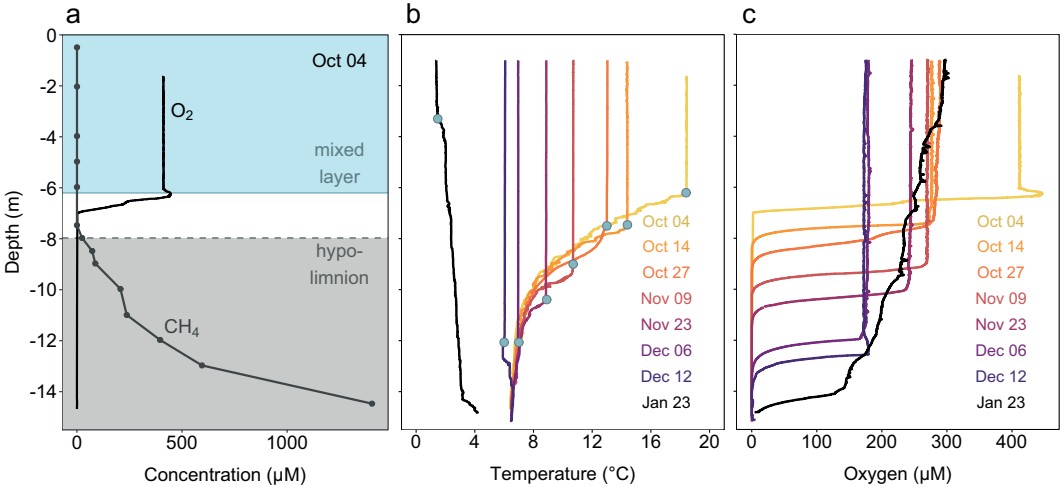

**Fig. 1 Stratification and progressive deepening of the mixed layer during lake overturn.** Depth profiles of physicochemical parameters in Rotsee during the lake overturn period of 2016/2017. **a** Stratified situation in Rotsee prior to lake overturn. Oxygen in the mixed layer and methane in the hypolimnion are separated from each other. The methane-enriched hypolimnion (≥2 µM) is shaded in grey, the mixed layer is shaded in blue. **b** Temperature profiles of the cooling lake in autumn and winter and the resulting expansion of the mixed layer at each sampling date. In January an inverse stratification established. Blue dots indicate the mixed layer depth. **c** Corresponding oxygen profiles at each sampling date. Colours in **b** and **c** indicate sampling date.

From October to December, a total of 4.2 Mg C of stored methane[14] gradually entered the expanding mixed layer in Rotsee. Nevertheless, median methane concentrations in the surface layer stayed low, ranging from 0.1–1.1 μM. During overturn, oxygen levels in the mixed layer dropped down to ~175 μM (Dec 12, Fig. 1c), likely due to methane and other reduced substances from the hypolimnion triggering abiotic and biotic oxygen consumption. From December to January, oxygen concentrations increased again.

**The methane-oxidizing assemblage prior to lake overturn**. We analyzed the MOB assemblage with a set of independent and mutually supportive methods. We used 16S rRNA gene sequencing to phylogenetically identify known groups of MOB, to estimate their proportion among the bacterial community and to assess MOB assemblage dynamics. The sequencing of *pmoA* mRNA and the quantification of *pmoA* transcripts with qPCR provided an independent assessment of the MOB assemblage as well as confirmation of transcriptional activity of the methane monooxygenase. To facilitate interpretation, we assigned the same colors to MOB ASVs (MOB-affiliated 16S rRNA Amplicon Sequence Variants) and aaASVs (*pmoA* amino acid Amplicon Sequence Variants) in Figs. 2 and 3 and supplementary figs., if they showed similar relative abundance and distribution pattern in a canonical correspondence analysis (Supplementary Fig. 2) and compatible placement within the phylogenetic trees (Supplementary Fig. 3).

On Oct 4, the proportion of MOB ASVs and the copy number of *pmoA* transcripts peaked at the upper boundary of the methane-enriched hypolimnion, although both measures showed that MOB were present throughout the water column (Fig. 2a, b). The hypolimnion harbored a higher proportion of MOB ASVs and more *pmoA* transcripts than the oxygen-rich mixed layer. Therefore, the hypolimnion represents a potential reservoir of MOB for the following lake overturn. The composition of the MOB assemblage differed strongly between the mixed layer and hypolimnion. The mixed layer was dominated by gammaproteobacterial type Ia MOB (Fig. 2a, b, Supplementary Fig. 4, blue) assigned to uncultivated groups CABC2E06/Lake cluster1 (ASV_5/aaASV1 and aaASV6). According to 16S rRNA gene-based analysis a *Methylocystis* (ASV_354, yellow) was also dominant; however, the corresponding aaASV7 *pmoA* transcript was rather underrepresented (Supplementary Fig. 4).

In the hypolimnion, other uncultivated type I MOB closely related to lacustrine *Crenothrix* prevailed (ASV_8, 35, 81, with 95.6–96.7% identity and aaASV2, 3, 8, with 96.1–97.4% identity to lacustrine *Crenothrix*; shown in shades of pink). In between the mixed layer and the methane-enriched hypolimnion the abundance of additional MOB sequences peaked, e.g. ASV_166 (Supplementary Fig. 4a) and aaASV12.

**MOB dynamics at the onset of the overturn**. By Oct 27 we observed an increase of MOB abundance and *pmoA* transcripts at the upper boundary of the methane-enriched hypolimnion (Fig. 2a, b). Although the mixed layer depth increased throughout October, it did not yet reach the methane-enriched hypolimnion (Fig. 2a, b). However, the deepening of the mixed layer brought oxygen-rich water closer to the methane-enriched hypolimnion.

By Nov 9 the mixed layer had finally reached and eroded a small part of the methane-enriched hypolimnion (Fig. 2a, b). The region occupied by the MOB peak on Oct 27 was replaced by CABC2E06 (ASV_5/aaASV1 and 6, blue), which already prevailed in the mixed layer in October and showed up again as the dominant MOB in the expanding mixed layer (Fig. 2a, b, Supplementary Fig. 4). On the other hand, *Crenothrix*-related MOB (ASV_8/aaASV2, pink)

remained restricted to the hypolimnion. From Nov 9 onwards, we no longer observed the MOB peak formation at the oxygen-methane interface, which is typical for the stable stratification period (Fig. 2a, b). Concurrently, the interface was constantly moved downwards as the mixed layer deepened.

**Dynamics of the MOB assemblage during the main phase of the overturn**. From Nov 9 to Dec 12 the mixed layer deepened from 9 to 12 m, thereby incorporating large amounts of stored methane. It must be assumed that the MOB assemblage from the hypolimnion was continuously transported into the mixed layer. However, the *Crenothrix*-related MOB that dominated there (Fig. 2a, b) were only detected at very low abundances in the mixed layer. Instead, a MOB with 97.2% rRNA gene fragment sequence identity to *Methylosoma difficile* (ASV_12/aaASV5, Supplementary Fig. 3a) was rapidly increasing in abundance, peaking at a median percentage of 16% of all bacterial sequences in the mixed layer on Dec 12 (Fig. 2a, b, orange). At this date, all MOB sequences together reached up to 28% of the bacterial sequences, with ASV_12 as the most important contributor to the increase in MOB abundance. This taxon was already present in October, but at very low proportion with a maximum relative abundance of 0.2% of bacterial rRNA gene sequences.

In January, an inverse stratification formed and reduced the depth of the mixed layer. MOB proportion and mRNA copy numbers of ASV_12/aaASV5 decreased and the MOB assemblage shifted again, with ASV_15/aaASV4 (*Methylobacter*, green, Fig. 2a, b) now dominating.

**Dynamics of potential methane-oxidation capacity and MOB cell numbers**. In October, type I MOB cell numbers as determined by CARD-FISH peaked at the upper boundary of the methane-enriched hypolimnion (Fig. 2c), corroborating the patterns observed with MOB ASVs and *pmoA* mRNA transcripts (Fig. 2a, b). Further, potential methane-oxidation rates showed a strikingly similar pattern (Fig. 2d). Even within the oxygen-depleted hypolimnion, potential methane-oxidation rates never fell below 0.3 μM d$^{-1}$ (Fig. 2d). Throughout the lake overturn, MOB cell numbers and potential methane-oxidation rates increased substantially within the mixed layer (Fig. 2c, d). By Dec 12, the potential methane-oxidation rates in the entire mixed layer exceeded the maximum rates observed during periods of stable stratification, when maximum rates are usually confined to a narrow layer at the oxygen-methane interface. The water volume capable of these high rates had thus expanded considerably (Fig. 2d). In combination, this led to an overall increase of the potential methane-oxidation capacity in the mixed layer from 0.01 Mg C d$^{-1}$ to 0.3 Mg C d$^{-1}$ between October and December (Fig. 2d) corresponding to a doubling of the rate every 12.7 days. At the same time, MOB cell numbers in the mixed layer increased with a net doubling time of ~9 days. After lake overturn was complete, i.e. between Dec 12 and Jan 23, potential methane-oxidation rates and MOB cell numbers generally decreased at the water depths formerly occupied by the mixed layer. The methane-oxidation rates and MOB abundance now increased with depth and peaked in the remaining methane-enriched bottom layer near the sediment (Fig. 2c–e). All measured MOB parameters were significantly correlated (Supplementary Table 1).

**Winners and losers: thriving mixed layer MOB and fate of hypolimnetic MOB**. The winners of the lake overturn were taxa of the mixed layer MOB assemblage (Fig. 3a), which contributed substantially to the increase of potential methane-oxidation rates (Fig. 3b). The profound shifts in the composition of the MOB

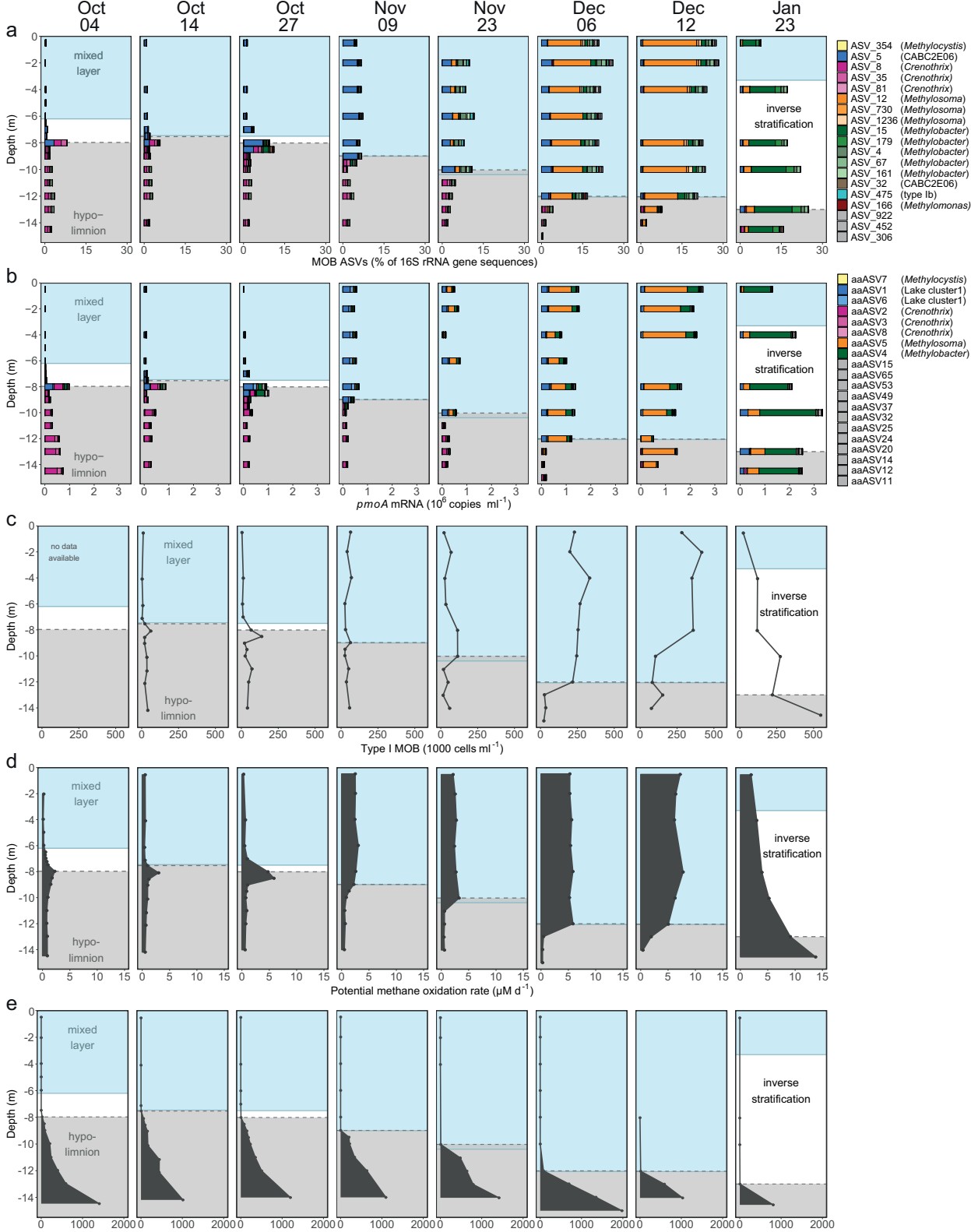

**Fig. 2 Depth distribution and dynamics of the MOB assemblage, *pmoA* transcripts, MOB cell numbers and potential methane-oxidation rates during lake overturn.** The mixed layer (blue background) depth increased with time, while the methane-rich hypolimnion starting at $\geq 2\,\mu M$ $CH_4$ (grey background) got gradually incorporated into the mixed layer. **a** Abundance and composition of MOB from 16S rRNA gene amplicon sequencing normalized to all bacterial 16S rRNA gene sequences. Colors distinguish MOB at the ASV level. **b** *pmoA* transcript abundance derived from qPCR quantification (copies ml$^{-1}$) multiplied with the relative abundance of *pmoA* mRNA aaASVs from amplicon sequencing. Colors correspond to aaASVs. ASVs and aaASVs which, based on multiple lines of evidence (see methods), are thought to originate from the same organism are encoded with the same color. **c** Type I MOB cell abundance as determined by CARD-FISH. **d** Potential methane-oxidation rates. **e** Methane concentration profiles.

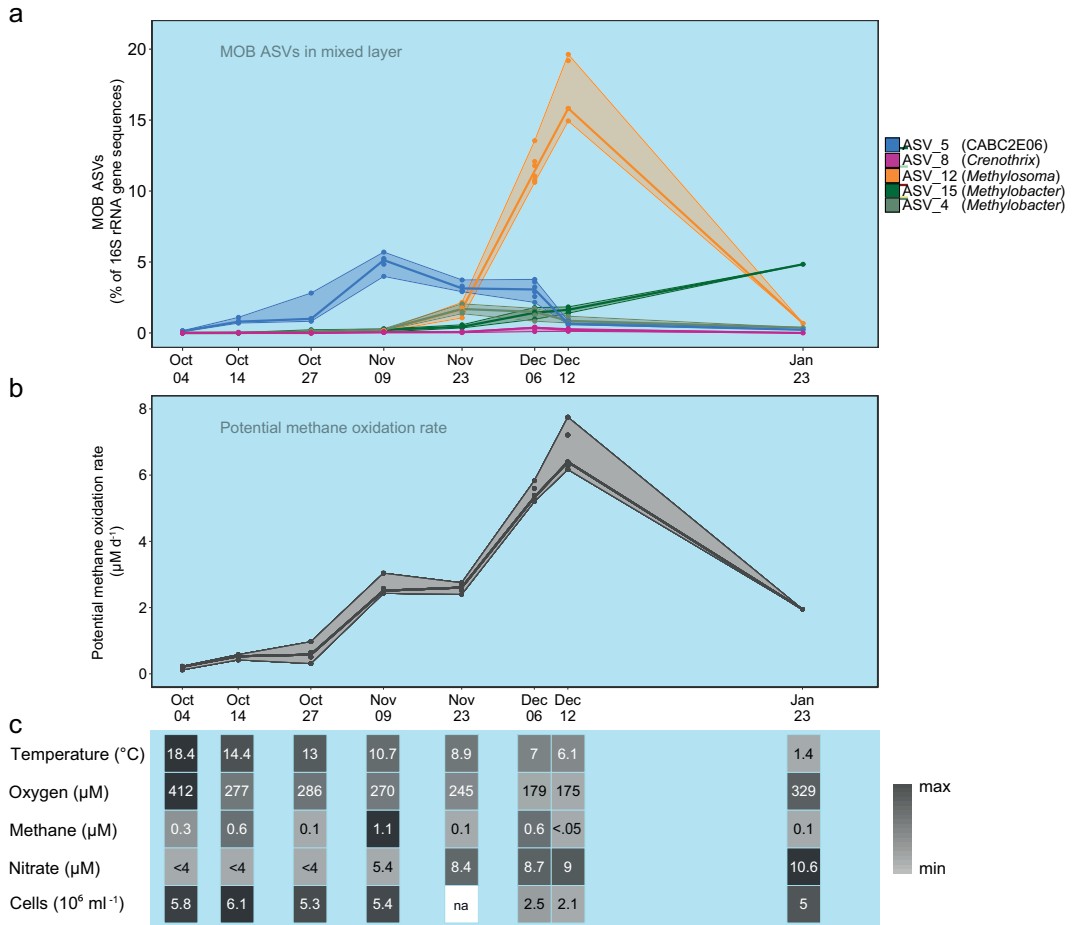

**Fig. 3 Mixed layer dynamics during lake overturn.** The minimum, maximum and median were derived from separately measured samples from different depths of the mixed layer. **a** MOB assemblage dynamics of the five most common ASVs within the expanding mixed layer during lake overturn. Maximum, minimum and median relative abundance of each MOB ASV within the mixed layer are visualized with lines. Colors correspond to MOB ASVs. **b** Dynamics of potential methane-oxidation rates during lake overturn. Lines depict maximum, minimum and median within the mixed layer. **c** Heatmap illustrating the corresponding change in physical, chemical and biological environmental parameters in the mixed layer over time. The median of the mixed layer samples is shown for each parameter and sampling date. Shades of grey are scaled between minimum (light grey) and maximum (black) values. Values below the limit of quantification are marked with "<" and missing data with "na". Raw data and sample size for **b** and **c** are provided in Supplementary Table 2.

assemblage occurred simultaneously with changes of physical and chemical conditions in the mixed layer (Fig. 3c).

In October and November CABC2E06 (ASV_5) dominated in the oxygen-rich mixed layer and increased in abundance at same time as the potential methane-oxidation increased from Oct 27 to Nov 09 (Fig. 3a, b). The early dominance of CABC2E06 (ASV_5) was followed by a pronounced increase of the *Methylosoma*-related ASV_12 until Dec 12, which led to dominance of ASV_12 during the lake overturn (orange, Fig. 3a). Prior to lake overturn ASV_12 was only detected at very low abundance. During dominance of ASV_12 the potential methane-oxidation rates peaked in the mixed layer (Fig. 3b). The growth phase of ASV_12 was accompanied by a drop in water temperature to below 10 °C, and increasing nutrient levels in the mixed layer (nitrate, Fig. 3c). ASV_12 peaked concurrently with the lowest measured oxygen concentration of 175 μM in the mixed layer during our study period (Fig. 3c). ASV_15, related to *Methylobacter tundripaludum* (98.6% identity) steadily increased during the later phase of the overturn and dominated the MOB assemblage on Jan 23 (Fig. 3a). At the same time lake water cooled to below 4 °C and oxygen levels replenished, while elevated nutrient levels persisted (Fig. 3c). The microbial cell numbers in the mixed layer were lower in December (Fig. 3c), but at the same time MOB percentage and absolute MOB cell numbers peaked (Figs. 2d and 3a).

The losers of the lake overturn, *Crenothrix*-related ASVs/aaASVs (ASV_8, 35, 81 pink), remained restricted to the eroding hypolimnion, until only a tiny fraction was left on Jan 23 (Fig. 2a, b, Supplementary Fig. 4). The *Crenothrix* were never able to establish in the mixed layer. They resided in water layers with high methane concentrations and low oxygen concentrations. A more flexible pattern was shown by *Methylobacter tundripaludum*-like ASV_4 (99.3% identity), which was present at low abundance in the hypolimnion and temporarily gained momentum in the mixed layer in November and December (Fig. 3a).

## Discussion

In this study, we provide, to our knowledge, the most comprehensive analysis so far of the response of a MOB assemblage to autumn lake overturn. We asked if the MOB assemblage grows fast enough to oxidize the stored methane before outgassing. Our results unambiguously show that this was the case: we documented a substantial MOB bloom during the lake overturn period (Fig. 2b, c) when MOB abundance increased until MOB represented up to 28% of the bacterial 16S rRNA gene sequences in the mixed layer (Fig. 2a). This growth in turn established a high methane-oxidation capacity in the expanding mixed layer (Fig. 2d). Detailed analysis revealed that potential methane-

oxidation capacity always exceeded the methane input from the hypolimnion and limited the emitted methane to a small percentage of the stored methane during lake overturn[14].

We further asked whether the standing MOB assemblage is activated, or a new assemblage successively takes over. We found that a temporal succession of MOB taxa underpins the increasing MOB abundance and methane-oxidation capacity (Fig. 3a, b). Together our results show that growth of a new MOB assemblage and its methane-oxidation capacity were the main mechanism effectively limiting outgassing of hypolimnion-stored methane during lake overturn.

Beyond these main findings our work provides a number of insights into the ecology of freshwater MOB. There is some evidence that the successional patterns of MOB, similar to vertical niche preferences discussed previously[16] are related to specific adaptations of the successful taxa. Dominance of ASV_5 (CABC2E06) occurring under, and being correlated with, elevated oxygen concentrations (Fig. 3, Supplementary Fig. 2) supports previous observations of this taxon being positively correlated with oxygen during stable stratification in four Swiss lakes[16]. Together, this supports the view that this MOB taxon thrives under high oxygen concentration and continuous but low-concentration methane supply. Currently, lack of a cultured representative or genome precludes further insights into the ecology of this prevalent taxon. The overturn winner ASV_12 (*Methylosoma*) was detected at low abundance in October in Rotsee prior to overturn and was not observed in three other Swiss lakes in a previous study[16]. This MOB taxon thus appears to be seasonally restricted or conditionally rare, but was the most spectacular profiteer of the lake overturn and highly relevant to methane mitigation during our study period (Fig. 3a). The only cultured representative of *Methylosoma*, *M. difficile* is micro-aerophilic[17], which is in line with our observation that ASV_12 peaked concurrently with the lowest measured oxygen concentrations in the mixed layer (Fig. 3c). The prevalence of ASV_15 (*Methylobacter*) in January could perhaps be explained by an adaptation of ASV_15 to low temperatures: Previous observations of psychrophily within the *Methylobacter* genus[18] provide some support for this hypothesis. *Crenothrix*-related MOB (ASV_8/aaASV2) prevalent in the hypolimnion were not able to establish in the mixed layer, although *Crenothrix* was presumably transported to the mixed layer continuously. The reason for this may be that *Crenothrix* is adapted to oxygen-deficient – high methane conditions, as proposed earlier[9,16]. Further, lacustrine *Crenothrix* may benefit from oxygen released from phytoplankton[19] or from temporally and spatially limited nano- to micromolar oxygen intrusions to the hypolimnion[20].

During the phase with the highest methane-oxidation capacity in the lake, the initial MOB assemblage of both, mixed layer and hypolimnion, had been almost entirely replaced by a new MOB assemblage (Fig. 2). Different MOB reached a high degree of dominance at different stages of the lake overturn, indicating that the changing environmental conditions favored temporal niche partitioning of MOB taxa (Fig. 3). It is notable, however, that despite the well-mixed situation and the often-observed dominance of a single taxon, a diverse MOB assemblage was nevertheless present throughout the overturn period. Answering the question whether the same or similar MOB are dominating the fall overturn every year will require a longer-term monitoring effort. Presence of MOB taxa with specific adaptation that allow them to take advantage of the rapidly changing conditions in the overturning lake is likely essential to establishing the methane-oxidation capacity that ultimately limits methane outgassing.

Although methane becomes available in the mixed layer, dilution and rapid oxidation keep concentrations low (maximum of 1.4 μM, Supplementary Tab. 2) as opposed to the situation in the hypolimnion (Fig. 2e). MOB in the mixed layer thus likely require a relatively high methane affinity for growth. We therefore speculate that the methane affinity of the mixed-layer MOB assemblage may be a critical factor for the amount of diffusive methane outgassing to the atmosphere during the overturn period. In addition to the inherent methane affinity of MOB taxa and assemblages, other traits like growth rates and ability to access nutrients likely affect the build-up of the methane-oxidation capacity in the lake and require further exploration.

The net doubling time of MOB in the expanding mixed layer was ~9 days. Nevertheless, MOB substantially increased their abundance and total potential methane-oxidation capacity in the mixed layer volume reached ~0.3 Mg carbon per day (Dec 12). While potential methane-oxidation rates may overestimate in situ rates under limiting methane concentrations, our oxidation capacity estimate is conservative as only potential methane oxidation to $CO_2$ was measured and incorporation into biomass is not included. Even though the MOB growth rates seem low compared to observations in MOB isolates[21,22] with doubling times of several hours, they may be in line with the environmental conditions: temperatures in the mixed layer dropped from 18.4 (Oct 4) to 6.1 °C (Dec 12), thus slowing process and growth rates are expected. Further, persistently low methane concentrations likely reduced effective growth rates. The doubling time given above represents a sum overall taxa. Because of the evident succession within the MOB assemblage, growth rates of e.g. taxa becoming dominant were certainly faster as other taxa stagnated or decreased in abundance at the same time (Fig. 3a). Finally, results presented here are net growth rates; gross growth rates are likely higher as mortality rates remain unknown.

While some estimates of global methane emissions from lakes assume that all stored methane is emitted to the atmosphere[3,23], our data are in line with a number of field studies that suggest a considerable proportion of stored methane is oxidized. One study estimated that 46% of the stored methane is emitted during autumn lake overturn[8], but most estimates are higher, claiming oxidation of 75–94% of the stored methane[6,7,24]. Rotsee methane emission data from eddy-covariance flux measurements and modelling[14] obtained in parallel with this study showed that in the present case even about 98% of the stored methane was oxidized. Our data demonstrate that this outcome was based on the robust response of MOB in the mixed layer where a succession of MOB maintained a high methane-oxidation capacity throughout the overturn.

We must therefore caution against adding methane storage to lake emission estimates. Certainly, lakes remain important sources of atmospheric methane: ebullition bypasses biological oxidation and rapid overturn events due to e.g. fast lake cooling and strong winds will lead to increased outgassing[6,14]. It does, however, appear reasonable to expect that processes as observed in Rotsee may occur in a considerable proportion of temperate lakes and might be important to global methane flux estimates. Most lakes in temperate regions are holomictic and have a mean lake depth of <25 m, similar to Rotsee. Further, small (size classes 0.1–1 and 1–10 km²), relatively shallow (average depth 5 and 5.4 m, respectively) lakes represent 28% of the estimated global lake area[25] and methane storage is frequently observed especially in small lakes[3]. The number of lakes with hypolimnetic methane storage may increase in future due to lack of recovery of lakes from eutrophication[26] and continuing eutrophication of other lakes[27], while global warming could strengthen summer stratification and, in turn, methane accumulation. A comprehensive understanding on the fate of stored methane in transitionally stratified lakes and information on the adaptability of MOB to different environmental conditions and their methane-oxidation kinetics are therefore critical.

In summary, we showed in this comprehensive study that successional changes in the MOB assemblage of the mixed layer of an overturning lake secured high MOB abundance with high methane-oxidation capacity that strongly reduced the atmospheric emissions of stored methane.

## Methods

**Environmental sampling and profiling during autumn overturn**. A temperate holomictic lake, Rotsee (47.072 N and 8.319 E) in Central Switzerland was sampled on 8 dates from Oct 4, 2016 to Jan 23, 2017 (see Fig. 1). Eutrophic lake Rotsee is situated at 419 m a.s.l., has a maximum depth of 16 m, is 2.5 km long and about 200 m wide[28]. Our sampling period covered the end of stratification with maximal methane accumulation in the hypolimnion and the subsequent autumn overturn. A moored thermistor chain with 1 m resolution provided temperature measurements every 3 h, documenting the mixing process and serving as a basis for scheduling the sampling dates. We defined the mixed layer depth as the part of the water column with a uniform temperature profile. At each sampling date, continuous profiles of temperature and oxygen (PSt1, Presens, Regensburg, Germany) were measured with the profiling in situ analyzer (PIA) as described previously[29]. Sampling depths for further analyses were set based on the PIA profiles. The PIA was equipped with a syringe rosette sampler (12 × 60 ml) for targeted depth sampling at 25 cm resolution, integrating about 5 cm water column, allowing for improved resolution compared to conventional Niskin bottle sampling, which integrates over >50 cm. Per depth six syringes were triggered to retrieve enough volume for different analyses.

Water samples for analysis of nitrate and dissolved inorganic carbon (DIC) were filtered with a 0.2 µm pore size cellulose acetate filter (Sartorius, Göttingen, Germany) and stored at 4 °C until analysis. Nitrate was measured with a flow-injection analyzer (SAN++, Skalar, Breda, The Netherlands). A total organic carbon analyzer (Shimadzu, Kyoto, Japan) was used for DIC quantification. Concentrations of DIC were taken into account when calculating potential methane-oxidation rates. Water samples for flow cytometry were fixed with formaldehyde and stained with SYBR Green I (Thermo Fisher Scientific, Waltham, MA, USA) at 37 °C for 15 min. Particles were counted on an Accuri C6 flow cytometer (BD Biosciences, San Jose, CA, USA) and microbial cell counts determined as described previously[30].

Samples for methane measurements were injected into airtight 40 ml serum vials filled with $N_2$ gas and 4 g NaOH pellets (≥98% purity, Sigma-Aldrich, Darmstadt, Germany) for preservation. The vials were prepared in the laboratory prior to fieldwork. Gas chromatography (6890N, Agilent Technologies, Santa Clara, CA, USA) was used to measure methane in the headspace. The GC was equipped with a Carboxen 1010 column (Supelco, Bellefonte, PA, USA) and a flame ionization detector. Control experiments showed that 4 g of solid NaOH released 32.6 nmol $CH_4$ on average which was accounted for when back-calculating methane concentrations in water based on Wiesenburg and Guinasso[31].

**Nucleic acid purification**. One hundred and ten millilitre of lake water were filtered on site through 0.2 µm cellulose acetate filters (Sartorius), which were instantly frozen on dry ice. Samples were stored at −80 °C until extraction. Genomic DNA and total RNA were extracted from the same filter with the AllPrep DNA/RNA Mini Kit (Qiagen, Hilden, Germany) with a bead beating step on a FastPrep-24 (MP Biomedicals, Santa Ana, CA, USA) and 150–212 µm glass beads. Residual DNA in RNA extracts was digested using the TURBO DNA-free kit (Thermo Fisher Scientific). DNA removal was checked by gel electrophoresis after PCR (16S rRNA gene, 27 f/1492r, 35 cycles) and if required, a second digestion was performed. If the second digestion was not sufficient to remove the DNA, the sample was excluded (one sample). cDNA was reverse transcribed from RNA with random hexamers using SuperScript IV First-Strand Synthesis System (Thermo Fisher Scientific). DNA and cDNA were used for qPCR and Illumina MiSeq sequencing.

**pmoA gene and transcript quantification**. The amount of *pmoA* DNA and mRNA copies in lake water during overturn was quantified by qPCR using the primer pair 189 f (3′-GGNGACTGGGACTTCTGG-5′) and mb661 (3′-CCGGM GCAACATGYCTTACC-5′)[32] on a LightCycler 480 (Roche Diagnostics, Rotkreuz, Switzerland). Ten microlitre reactions were performed with 0.2 µM primer, 2 µl of 1:10 diluted DNA or cDNA and the SYBR Green I Master mix (Roche). We used PCR conditions adapted from Henneberger et al.[33] with 10 min initial denaturation, 15 s denaturation during cycling and acquisition at 79 °C. Standards were obtained by 10-fold serial dilution of *pmoA* containing plasmid, which were measured in quadruplicates. Sample assays were performed in triplicates and analysed with the LightCycler 480 software (v1.5.1.62) with the second derivative maximum method.

**16S rRNA and pmoA sequencing and analysis**. Bacterial 16S rRNA gene and rRNA, amplified using primers S-D-Bact-0341-b-S-17 (3′-CCTACGGGNGGCW GCAG-5′) and D-Bact-0785-a-A-21 (3′-GACTACHVGGGTATCTAATCC-5′)[34] and functional marker *pmoA* gene and transcript (189f, mb661)[32] amplicon

libraries were sequenced on the Illumina MiSeq platform (Illumina Inc., San Diego, CA, USA). Briefly, the first PCR was performed in triplicate with tailed forward and reverse primers using NEBNext Q5® Hot Start HiFi PCR Master Mix (New England BioLabs, Hitchin, UK) and 17 and 25 cycles for the bacterial and functional marker, respectively. Illumina barcodes and adapters were attached to pooled and purified products in a second PCR (8 cycles) with the Nextera XT Index Kit A and D (Illumina Inc.). Libraries were purified with Agencourt AMPure XP kit (Beckman coulter, Indianapolis, IN, USA), quantified (Qubit, Thermo Fischer Scientific) and pooled equimolarly. After quality check on a Tape Station 2200 (Agilent Genomics, USA) the two sequencing runs with 600-cycle MiSeq reagent kit v3 and 10% PhiX were performed at the Genetic Diversity Centre (ETH Zurich).

Amplicon sequence variants (ASVs) were inferred with the DADA2 pipeline[35] (v1.6.0) in R[36] (v3.4.1). 16S rRNA reads were trimmed to 270 nt (forward) and 210 nt (reverse) and *pmoA* reverse reads were trimmed to 235 nt. At the first instance of a quality score (Q) of less or equal to two the read was truncated and reads with ambiguous bases or an expected error rate $EE = \sum 10^{\left(-\frac{Q}{10}\right)}$ exceeding three were removed. After calculating error rates the reads were dereplicated and denoised to infer exact sequence variants. After merging forward and reverse reads chimera were removed. Samples with less than 5000 reads were removed. Taxonomy of 16S rRNA ASVs was assigned using the SILVA reference database (v132)[37] and reads assigned to mitochondria or chloroplasts were removed with phyloseq[38] (1.24.2) in R (3.5.0) and the read counts were transformed to relative abundances. The 16S rRNA dataset was screened for known methanotrophic groups[39] within the order *Methylococcales*, genera within class *Alphaproteobacteria* and phylum *Verrucomicrobia*, and *Candidatus* Methylomirabilis. ASVs reaching >0.2% relative abundance in at least three samples were retained for further analysis. To increase readability of Fig. 2 two samples (Oct 04; 7.25 m and 6.75 m) were omitted from the figure, which did not affect data interpretation. Raw data for these samples are available from data repositories (see Data availability). Amino acid sequences (aaASV) were derived from *pmoA* ASVs in MEGA7[40] and were retained for further analyses when exceeding 2% relative abundance in at least one sample. Alignment (Muscle) and neighbor-joining trees were constructed in MEGA7 with 10000 bootstrap replications using Jukes-Cantor evolutionary distance for 16S rRNA sequences or Poisson correction method for *pmoA* amino acid sequences. To facilitate interpretation, we assigned the same colors for MOB ASVs (MOB-affiliated 16S rRNA amplicon sequence variants) and aaASVs (*pmoA* amino acid operational taxonomic units) if they showed similar relative abundance and distribution patterns as observed in a canonical correspondence analysis (Supplementary Fig. 2), and if placement in the phylogenetic trees was compatible with this interpretation.

**Potential methane-oxidation rates**. Water samples were filled into 60 ml serum vials in the field and laboratory incubations were started the same day. The applied procedure to determine potential methane oxidation was adapted from Oswald et al.[19]. Each sample was purged with $N_2$ to remove residual methane, amended with non-limiting concentrations of oxygen (~50 µmol L$^{-1}$) and $^{13}C$-$CH_4$ (~80 µmol L$^{-1}$), and subdivided into five exetainers[41] (6 ml). The incubation temperature for each date was chosen based on the in situ temperature profile at the date of sampling. For the first three dates a temperature between the mixed layer temperature and the bottom temperature (~6 °C) was chosen, for later dates the mixed layer temperature was chosen (incubation temperatures from Oct 4 to Jan 23 in °C: 13, 11, 11, 11, 8.9, 7, 6, 4). Incubations were performed in the dark and stopped after 0, 3, 13, 25 and 50 h with 100 µl ZnCl$_2$ 50% w/v. The increase of $^{13}C$-$CO_2$ was determined with GC-IRMS (IsoPrime, Micromass, Wilmslow, UK). Rates were derived from a linear regression of the $^{13}C$-$CO_2$ production over time, taking into account the background DIC in the sample. Carrara marble (ETH Zurich, δ$^{13}C$ of 2.1‰) served as a standard.

**MOB quantification with CARD-FISH and calculation of doubling times**. To quantify MOB dynamics in terms of cell numbers we applied CARD-FISH. Samples were fixed with formaldehyde (2% final concentration) for 3–6 h on ice prior to filtration onto 0.2 µm polycarbonate nucleopore track-etched membrane filters (Whatman, Maidstone, UK). Dry filters were stored at −20 °C until further processing. Filters were embedded in 0.2% low gelling point agarose (Metaphor, Lonza), cells were permeabilized with lysozyme (10 mg ml$^{-1}$) for 70 min at 37 °C and peroxidases inactivated in 10 mM HCl for 10 min. The horseradish peroxidase labelled probes Mg84 (5′-CCACTCGTCAGCGCCCGA-3′), Mg669 (5′-GCTACA CCTGAAATTCCACTC-3′) and Mg705[42] (5′-CTGGTGTTCCTTCAGATC-3′) were mixed to target known type I MOB. As control probes EUB338I-III (universal bacterial probe, positive control) and NON338 (negative control)[43,44] were separately applied to each sample. Filters were incubated at 46 °C for 2.5 h with 150 µl of the respective hybridization buffer (20% formamide Mg-mix and 35% for EUB and NON338) containing 1 µl of probe (50 ng µl$^{-1}$). After washing, filters were placed into the amplification buffer containing tyramide labelled with Oregon Green 488 at 37 °C for 30 min in the dark. DAPI was used for counterstaining and the filters were mounted on glass slides in a mix of Citifluor and Vectashield (4:1).

Epifluorescence images were taken with an inverted microscope (DMI6000 B, Leica, Wetzlar, Germany) and a ×100/1.3 objective (PL-FL, Leica). To obtain clear images, multifocal images were recorded and merged with LASX software (Leica). DAPI and the corresponding CARD-FISH images of 22 randomly

selected fields per filter were used for semi-automated cell counting with the daime (v 2.0) software[45]. In daime, image histograms were adjusted manually; objects were detected with the edge detection algorithm and split with a watershed segmentation-based algorithm. Object detection was improved manually if necessary. An exemplary cell identification with daime is shown in Supplementary Fig. 5.

To calculate doubling times the median of MOB cell numbers in the mixed layer multiplied by the mixed layer volume were used to calculate growth rates (r) using the natural logarithm plotted against time (Oct—Dec 12). Doubling times were derived as $dt = \ln(2)/r$ (ref. [46]). The doubling of the methane-oxidation capacity in the mixed layer was calculated analogously.

**Statistics and reproducibility**. Canonical correspondence analysis (CCA) was performed with phyloseq[38] in R on the MOB ASV relative abundance among bacterial 16S rRNA gene sequences and the aaASV percentage of pmoA mRNA sequences. We used a Chi-square dissimilarity matrix and scaled physicochemical parameters ($O_2$ and $CH_4$ concentrations and temperature) as constraints. Constrained axes were tested for significance with anova.cca of vegan v2.5.2.

Spearman rank correlation between MOB parameters and potential methane-oxidation rate were calculated in R and p values were adjusted for multiple comparisons using the Benjamini & Hochberg (BH) method.

The methane-oxidation capacity was calculated based on the median potential methane-oxidation rate in the mixed layer multiplied by the volume of the mixed layer.

**Reporting summary**. Further information on research design is available in the Nature Research Reporting Summary linked to this article.

## Data availability
The sequencing files have been submitted to the European nucleotide archive under the project number PRJEB32413. All other data (environmental variables, qPCR-based gene and transcript abundances and CARD-FISH-based cell abundances as tables in comma separated values format and pmoA aaASV and 16S rRNA ASV sequences in FASTA format) are available from the Eawag Research Data Institutional Collection[47].

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

## Acknowledgements

We thank Carsten Schubert and Serge Robert for access to the GC-IRMS and support with methane-oxidation rate measurements and Carsten Schubert for his support, review of the manuscript, and helpful discussions. We thank Karin Beck, Patrick Kathriner, Miro Meyer and Michael Plüss for assistance in the field and laboratory. Sequencing data generation and analysis were done in collaboration with the Genetic Diversity Centre (GDC), ETH Zurich. The research was funded by Swiss National Science Foundation (grant CR23I3_156759).

## Author contributions

M.M. was responsible for carrying out the research, performed laboratory and data analysis and wrote the original manuscript. M.Z. and J.D. contributed methane-oxidation rate measurements and J.D. contributed CARD-FISH analyses. M.M., M.Z., J.D. and A.B. performed fieldwork. H.B., B.W. and M.Z. contributed to writing, reviewing and editing the manuscript. H.B., B.W. and A.B. conceptualized and supervised the research, acquired funding and provided the research infrastructure. M.M. and M.Z. designed the study.

## Competing interests

The authors declare no competing interests.
