## [Peer Review File · Communications Biology]

Reviewers' comments:

Reviewer #1 (Remarks to the Author):

This study investigates the fate of the methane previously confined in the hypolimnion of stratified lakes during seasonal turnovers. Using 16S rRNA and pmoA mRNA sequencing, qPCR, CARD-FISH and measurements of methane oxidation rates, the authors of this manuscript describe the succession of methanotrophic communities and associated activities depending on lake depth and turnover progression in the Lake Rotsee. They have highlighted a succession of different lineages of methane-oxidizing bacteria in the mixed layer of the lake and an important increase of the potential methane oxidation rate in the mixed layer during the turnover of the lake. Their results suggest that a large part of the methane confined in the hypolimnion during the stratified season are efficiently consumed by methane oxidizing bacteria in the mixed layer during seasonal overturns avoiding release in the atmosphere of large amounts of this potent greenhouse gas.

This is a very interesting work and I really appreciate to read this study and these valuable results. This study attempts to answer to a clear and important research question : is the methane stocked in the hypolimnion of lakes totally released in the atmosphere when lakes seasonally overturn? The question is properly answered with the methodology used even if studies on more lakes are always of interest. The study lacks of replicates but the design sounds appropriate. The novelty of the study is obvious and this study is of interest for everything working on freshwater. However, I have some concerns about the manuscript that I describe in details below (major and minor comments). My main concerns are (1) some conclusions are disconnected from the results and could give the impression to over-interpretation of the results. Each conclusion has to be clearly linked to the results on which it is based to avoid this impression of overselling. See below for more details. (2) Some of the results are interpreted in the result section and I think it would be valuable to reorganized the manuscript to clearly separate raw results from interpreted results and interpretation. (3) Some details lack in the method section to really understand what the authors exactly did.

MAJOR COMMENTS

RESULTS

l. 80 Does this quantity of carbon represent the amount of carbon which enters in the mixed layer of the whole lake? Are there differences between the different parts of the lake?

l. 157-160 I am not sure I understand how the results presented in this section have been obtained. Did the authors consider a constant and regular increase of the methane oxidation rates between October and December? How has the "9 days" of doubling time been calculated? Did the authors consider that the same variation in microbial communities and in methane oxidation rate occur over the whole surface / the whole mixed layer of the lake?

l. 193 Is the lake covered with ice and snow during winter? If it is, it could be interesting to discuss the impact of this parameter on the microbial community structure.

l. 119, 122-131, 151, 167-206 These parts of the result section contain some interpretations of the results. I think these interpretations would be better in the discussion part.

DISCUSSION

l. 210-212 and 218 Some of the claims presenting in this section lack of a reminder of some results. I think the authors have to more used their results to explain why they could conclude that MOB assemblage was reacting fast enough to minimize outgassing before conclude it in that way.

Otherwise, this could be perceive as over-interpretation of the results.

METHODS

l. 292 As described in the method section, analyses of ammonium and DIC were performed. However, I could not find any result in the manuscript about these analyses. These analyses may either totally

be removed if not used or corresponding results have to be presented and discussed in the manuscript.

l. 296-299 As described in the method section, some water samples were collected for flow cytometry analyses. However, I could not find any result in the manuscript about these analyses with the exception of the raw data in the Supplementary Table 2. Maybe the authors could present and discuss these results in the manuscript (or removed if not used at all).

l. 288 (+ l. 292, 296, 300 and 308) This method of sampling seems particularly interesting. I read in Kirf et al. 2014 (ref. 27) that the PIA was equipped with a carousel syringe sampler with 12 x 60 mL syringes. Did the authors use this carousel in their own study? I have some difficulties to understand how the sampling has been performed. Did the authors use one syringe of 60 mL per depth to complete chemical and microbiological analyses (ammonium, nitrate, DIC, cytometry, methane, nucleic acid purification)? Did the authors collect 110 mL of lake water (l. 308) by putting the PIA several times in the water? Did the authors use a larger carousel to collect the 16 samples for the first date of sampling (Oct 04)? Please add details in the manuscript.

l. 308 A volume of 110 mL of water may sound a little bit low to cover the entire microbial communities inhabiting the lake. Could the authors explain this choice of 110 mL?

l. 321 Have the PCR1 triplicate been sequenced separately? As I understand it, the authors performed only technical replicates ("the first PCR was performed in triplicate", l. 321) but no sampling replicates (replicates for the same depth of the lake at the same date, sequenced separately). It might have been interesting to perform this kind of sampling replicates to know if the observed microbial communities effectively reflect a true microbial communities in the lake. That being said, the congruence between the different sampling depth is, to a certain extent, convincing. Could the authors explain why they chose to not performed sampling replicates and to not collect water in another place in the same lake to have a better overall picture of what is happening in the whole lake?

l. 376 How did the authors deal with autofluorescence of suspended particles (for CARD-FISH and flow cytometry experiments)?

FIGURES

Figure 3A Did the maximum, minimum and median relative abundance represent the different values obtained from the technical triplicates? If it is, I am not sure this statistics are really appropriate in this kind of graph.

MINOR COMMENTS

ABSTRACT

l. 22 It could be interesting to add a summary of the method used in this study to allow the reader to know what the authors did (16S rRNA gene and pmoA mRNA amplicon sequencing, qPCR, CARD-FISH, potential methane oxidation rate...)

RESULTS

l. 81 Since this article focuses on the methane consumption along the depth of the lake, I suggest that the supplementary figure 2 be part of the main figures.

l. 97-106 I agree with the interest to keep the Figure 2 as the main figure compared with the Supplementary Figure 5 to better see the variations of MOB proportion depending on depth and sampling date. However, since the data for the mixed layer are unreadable in the Figure 2, I suggest to also refer in this section to the Supplementary Figure 5 which represents the proportion of the different MOB lineages among all identified MOB.

l. 145 Please add the taxonomic affiliation for ASV_15 as mentioned later in the text (*Methylobacter tundipaludum*, l. 191)

l. 148-150 Please refer to Figure 2B and 2C.

l. 150 Please refer to Figure 2D

l. 152 Please change (Fig. 2C) to (Fig 2D)

DISCUSSION

I. 218 Please change (Zimmermann et al 2019) to the reference number 14.

METHODS

I. 280 Please add some characteristics of the lake : maximum or average depth, length, width... This allows easier comparison with other lakes

I.300 Did the samples for methane measurements filled with N₂ gas and NaOH pellets directly on the field or did the vials be prepared before going to the field?

I. 327 Have the libraries been pooled equimolarly?

I. 333 How many sequences were used to build the ASV table? Have the number of sequences been normalized at a given number of sequences per sample?

I. 343 Which type of alignment has been used to build the phylogenetic trees?

I. 376 Which fluorophores have been used of the CARD-FISH experiments?

I. 384 Maybe the authors should add some CARD-FISH pictures in supplementary data to show which type of marked cells was counted with Daime.

FIGURES

Figure 2 I suggest to add affiliations in the caption. For example, ASV_5 / CAB2E06 and ASV_354 / Methylocystis... This would greatly facilitate reading.

Figure 3A As I understand, the Figure 3A represent another way to visualize the data presenting in the Figure 2A. For greater clarity, the authors could only put on this graph the main MOB lineages (ASV_5, ASV_12, ASV_15, ASV_8, ASV_4). The other MOB lineages are unreadable on this graph and it is thus difficult to read.

Reviewer #2 (Remarks to the Author):

The manuscript by Mayr et al. describes the changes in the community composition and CH₄ oxidation capacity of the methanotrophic community inhabiting lake Rotsee during the autumn overturn period. The manuscript is well written and analyses appropriate. However, the authors are overselling a bit the story and it's novelty, and downgrading the value of earlier studies. Thus, some of the statements about the novelty of the study should be toned down. To the discussion should be added how realistic the measured CH₄ oxidation rates are compared to the conditions in the nature.

MINOR COMMENTS

L25-27 I would not say that this is a "previously unknown" phenomenon. For example, Taipale et al (AME 2011), which is cited elsewhere in the ms, already stated that methanotroph abundance was especially high during autumn overturn.

L37-39 Could you please provide some citation to this? I'm only aware of literature stating that carbon transfer from CH₄ to biomass in oxic conditions is an important energy source for the lake food web (e.g. Jones & Grey 2011, Freshwater Biology) and reports of relatively high C uptake to biomass (e.g. He et al. 2015, Environmental Microbiology).

L274-277 Since the amount of oxidized CH₄ is in the same range as in previous studies (L256), "extraordinarily" is quite strong term. Also, it suggests that lake here would not be representative, which would mean that no generalization should be done based on this study. In my opinion "high CH₄ oxidation capacity" would be strong enough statement.

L331 You mean the reads were cut at the position where the quality score dropped below 2?

L332 Exceeding three what? %? Bases?

L332 Were the forward and reverse reads assembled or analysed separate?

L350-358 This part would make more sense to be before 16S and pmoA analysis section, since the data processing from this material is explained there.

L364 Please add how close to in situ temperature the incubations were done and to which direction the deviation was.

Reviewer #3 (Remarks to the Author):

In the manuscript "Growth and rapid succession of methanotrophs effectively limit methane release during lake overturn" the authors report their findings on the change in community structure and abundance of methane-oxidizing bacteria (MOB) as well as potential methane oxidation activity along the depth profile of Lake Rotsee during the annual lake overturn.

As mixing proceeds, the community of MOB is changing and minor players during the stratified period outgrow the other MOB. The increase in MOB abundance results in a significantly increased potential methane oxidation capacity in the lake and limits methane release into the atmosphere.

This study nicely combines complementary methods to support the findings. 16S rRNA gene-based methods are complemented by pmoA mRNA-based quantification and CARD-FISH and backed up with potential methane oxidation rate measurements and nutrient data. Together, these data combine to a compelling picture of the dynamics in MOB community during a gradual mixing event. Although the study was limited to one lake, it clearly shows that methane stored in the anoxic hypolimnion of lakes is not released completely to the atmosphere upon mixing. In addition, the study emphasizes that MOB communities can be very dynamic and that different MOB clearly occupy different niches! These findings are clearly interesting to the community.

I only have a few specific comments:

- 1) In the color printout I used to review the paper, the different shades of gray in Fig 1B and C are very hard to distinguish, especially the darker shades. I suggest using additional color to increase the contrast between the different lines - especially since many figures already use color.
- 2) The grey shading in Fig. 3C makes the numbers really hard to read in my printout. I suggest changing the font to bold to increase the contrast or skip the shading. In my opinion, the shading does not add additional information as the table is short enough to actually read the numbers.
- 3) The sampling year in the main text and the methods is contradicting. Sometimes it's 2016, sometimes 2017.
- 4) Just a suggestion: Would it be possible to calculate a maximum growth rate of the MOB community using growth yield data from the literature and methane flux data you can calculate from your measurements? I would be interested to see whether the calculated max growth rates based on the amount of methane mixed into the mixed layer roughly match the observed rates. Would be a nice addition if they do, but would only distract from the main message of the paper if they do not.
- 5) I was not aware of methane released by dissolving NaOH pellets. Thanks for adding this information!
- 6) The flow cytometry data mentioned in the methods are not in the main article, but seem to fit well together with the type1 MOB CARD-FISH and 16S gene copy number data. Is there a reason not to report relative CARD-FISH/particles numbers as you did with the MOB-general 16S ratio?

Please find below our detailed replies to the comments of the three reviewers. We provide a summary of the main changes made to address the key concerns. Point-by-point replies to all reviewer remarks are included below.

A manuscript with changes highlighted using Word's track changes function is provided as a separate file.

Summary of main changes:

We improved the link between claims made in the Discussion section and our Results, which was a main concern of reviewer 1 and the editor. To do so, we added or changed formulations throughout the Discussion section and added more cross-references to the figures. Several comments concerned lack of details in our Methods section. In response, the Methods section has been amended with several methodological details to clarify all points raised by the reviewers. Also, we added an illustrative figure to the Supplementary Material that explains how MOB cells were counted with daime (Supplementary Fig. 5). Further, we checked that all data types suggested by our Methods section are actually reported in Results. In some cases we clarified in which analyses the data was incorporated. The readability of Figure 1B, C and Figure 3A, C was changed according to the suggestions of the reviewers. The methane concentration profile from the Supplementary Material was moved to Figure 2 as Figure 2E. We have toned down some statements to avoid the criticism of overselling our study in the Abstract, Introduction and Discussion (see detailed response below). Interpretations made in Results were moved to a separate paragraph in the Discussion for a better separation of results and interpretation. Further, we have addressed all minor comments raised by the reviewers, which led to small changes in all sections of the manuscript (see detailed response).

Detailed Response:

We thank the three reviewers for their careful and thoughtful assessment – we have worked on all points raised by the reviewers and provide detailed replies and descriptions of the changes made to the manuscript in the detailed response below. Line numbers refer to the revised manuscript.

Reviewer #1 (Remarks to the Author):

This study investigates the fate of the methane previously confined in the hypolimnion of stratified lakes during seasonal turnovers. Using 16S rRNA and pmoA mRNA sequencing, qPCR, CARD-FISH and measurements of methane oxidation rates, the authors of this manuscript describe the succession of methanotrophic communities and associated activities depending on lake depth and turnover progression in the Lake Rotsee. They have highlighted a succession of different lineages of methane-oxidizing bacteria in the mixed layer of the lake and an important increase of the potential methane oxidation rate in the mixed layer during the turnover of the lake. Their results suggest that a large part of the methane confined in the hypolimnion during the stratified season are efficiently consumed by methane oxidizing bacteria in the mixed layer during seasonal overturns avoiding release in the atmosphere of large amounts of this potent greenhouse gas.

This is a very interesting work and I really appreciate to read this study and these valuable results. This study attempts to answer to a clear and important research question : is the methane stocked in the hypolimnion of lakes totally released in the atmosphere when lakes seasonally overturn? The question is properly answered with the methodology used even if studies on more lakes are always of interest. The study lacks of replicates but the design sounds appropriate. The novelty of the study is obvious and this study is of interest for everything working on freshwater.

However, I have some concerns about the manuscript that I describe in details below (major and minor comments). My main concerns are (1) some conclusions are disconnected from the results and could give the impression to over-interpretation of the results. Each conclusion has to be clearly linked to the results on which it is based to avoid this impression of overselling. See below for more details. (2) Some of the results are interpreted in the result section and I think it would be valuable to reorganized the manuscript to clearly separate raw results from interpreted results and interpretation. (3) Some details lack in the method section to really understand what the authors exactly did.

Answer (1) The section in the discussion listed in the specific comment has been revised – we tried to keep a balance between not restating results too extensively yet more clearly linking our interpretation and main claims with the results achieved in this paper and elsewhere.

We have also checked the discussion section throughout. While we did not notice any further instances of unsubstantiated interpretations, we have nevertheless edited some parts and added additional internal references to our results for greater clarity. Finally, we toned down our claim of novelty in the first line of the discussion, from “first” to “most comprehensive so far” (Line 190) to avoid the allegations of overselling and to acknowledge the contribution of Taipale 2011, which is now also pointed out in the introduction (Line 53-55, see also replies to reviewer 2).

Answer (2) We removed interpretation of results in the results section and reorganized the manuscript accordingly (see also answer below) to separate raw results from interpretation and moved important points to a separate paragraph in the discussion section.

Answer (3) We added the required details to the methods section (see also answers below).

MAJOR COMMENTS

RESULTS

l. 80 Does this quantity of carbon represent the amount of carbon which enters in the mixed layer of the whole lake? Are there differences between the different parts of the lake?

Answer: Yes, this quantity is the total amount of methane that entered the mixed layer in Rotsee during overturn. We have clarified in the text that this refers to the lake as a whole. Line 83-84: "From October to December, a total of 4.2 Mg C of stored methane¹⁴ gradually entered the expanding mixed layer in Rotsee." We calculated the amount of methane neglecting potential horizontal variability. Amounts of methane were calculated considering the methane profile and the bathymetry of the lake. Because horizontal mixing in lake is usually very strong we do not expect major differences between different parts of the lake.

l. 157-160 I am not sure I understand how the results presented in this section have been obtained. Did the authors consider a constant and regular increase of the methane oxidation rates between October and December?

How has the "9 days" of doubling time been calculated? Did the authors consider that the same variation in microbial communities and in methane oxidation rate occur over the whole surface / the whole mixed layer of the lake?

Answer: We assumed an exponential increase of both the rate and the MOB cells in the mixed layer from October to December. The median cell number (or rate) of the mixed layer was multiplied by the mixed layer volume at the respective date and the natural logarithm was plotted against time. The doubling times were then calculated as $dt = \ln(2)/r$ (ref. 46) where r is the slope of the equation. This assumes an exponential increase of MOB cells and methane oxidation rate within the mixed layer from October to December ($R^2=0.94$ MOB cells, $R^2=0.96$ for the methane oxidation rates). The given doubling times give an estimate of the net growth rates averaged over the growing phase in the mixed layer.

We described the calculation in the methods section but it was somewhat buried among other things. Therefore we increased visibility by changing the header: Line 387: "MOB quantification with CARD-FISH and calculation of doubling times" and we slightly changed the description: Line 409-4122: "To calculate doubling times the median of MOB cell numbers in the mixed layer multiplied by the mixed layer volume were used to calculate growth rates (r) using the natural logarithm plotted against time (Oct – Dec 12). Doubling times were derived as $dt = \ln(2)/r$ (ref. 46). The doubling of the methane oxidation capacity in the mixed layer was calculated analogously."

l. 193 Is the lake covered with ice and snow during winter? If it is, it could be interesting to discuss the impact of this parameter on the microbial community structure.

Answer: Lake Rotsee is only very rarely covered with ice. Last time with considerable ice was in 2012 (according to the news - which was the first time since 26 years). I agree it is very interesting to investigate microbes under the ice, but Rotsee is not suitable for this kind of study, which is why we do not discuss this in the paper.

l. 119, 122-131, 151, 167-206 These parts of the result section contain some interpretations of the results. I think these interpretations would be better in the discussion part.

Answer: Yes, thank you - we checked the results section and removed interpretations from the results and moved them to the discussion. We added an additional paragraph to the discussion section, where these interpretations can now be found.

DISCUSSION

l. 210-212 and 218 Some of the claims presenting in this section lack of a reminder of some results. I think the authors have to more used their results to explain why they could conclude that MOB assemblage was reacting fast enough to minimize outgassing before conclude it in that way. Otherwise, this could be perceive as over-interpretation of the results.

Answer: We have revised this section (Line 190-203) in order to better link our interpretation and main claims with the results achieved in this work and work of others, while trying to avoid extensive

restating of the results. In particular, regarding the speed of the MOB response, we now more clearly reference the considerable work that was done to address this question in a separate paper (Line 196-198). We have edited also some other parts of the discussion and added additional internal references for greater clarity. Finally, we toned down our claim of novelty in the first line of the discussion, from “first” to “most comprehensive so far” in order to avoid the allegations of overselling and to acknowledge the contribution of previous work, which was also changed in the introduction. (Line 53-55 as requested also by reviewer 2).

METHODS

l. 292 As described in the method section, analyses of ammonium and DIC were performed. However, I could not find any result in the manuscript about these analyses. These analyses may either totally be removed if not used or corresponding results have to be presented and discussed in the manuscript.

Answer: We removed ammonium from the methods section, as the data is not shown in the manuscript. We keep DIC because the DIC concentration is important for calculating the methane oxidation rates and we clarified this in the text. Line 305-306: “Concentrations of DIC were taken into account when calculating potential methane oxidation rates.”

l. 296-299 As described in the method section, some water samples were collected for flow cytometry analyses. However, I could not find any result in the manuscript about these analyses with the exception of the raw data in the Supplementary Table 2. Maybe the authors could present and discuss these results in the manuscript (or removed if not used at all).

Answer: The flow cytometric cell numbers are shown in the table of Figure 3C to give some background information on the environment and to roughly put the 16S rRNA gene relative abundances into context. In addition, the cell numbers were used to inform us of how much volume to filter on the filters for CARD-FISH microscopy. Therefore, we would like to keep this in the method section.

We highlighted the microbial cell numbers in the revised results section with one sentence Line 180-181 “The microbial cell numbers in the mixed layer were lower in December (Fig. 3C), but at the same time MOB percentage and absolute MOB cell numbers peaked (Fig. 3A, Fig. 2D).”

l. 288 (+ l. 292, 296, 300 and 308) This method of sampling seems particularly interesting. I read in Kirf et al. 2014 (ref. 27) that the PIA was equipped with a carousel syringe sampler with 12 x 60 mL syringes. Did the authors use this carousel in their own study? I have some difficulties to understand how the sampling has been performed. Did the authors used one syringe of 60 mL per depth to complete chemical and microbiological analyses (ammonium, nitrate, DIC, cytometry, methane, nucleic acid purification)? Did the authors collect 110 mL of lake water (l. 308) by putting the PIA several times in the water? Did the authors use a larger carousel to collect the 16 samples for the first date of sampling (Oct 04)? Please add details in the manuscript.

Answer: Yes, in our study we used the same rosette sampler (12x60ml) as Kirf et al. 2014. Here is a small scheme (Fig. 1) of the sampling design, which we developed in order to have enough volume for all the different analyses. We made 2 DNA/RNA filters (one back-up filter) with 2 syringes each and two more syringes for methane/flow cytometry/CARD-FISH and the 13C-CH4 incubations. Like this we were able to sample two different depths with one cast.

Figure 1 Sampling design with the syringe rosette sampler PIA.

We clarified the sampling strategy by adding more information to the methods: Line 295-301: “At each sampling date, continuous profiles of temperature and oxygen (PSt1, Presens, Regensburg, Germany) were measured with the profiling in-situ analyzer (PIA) as described previously²⁹. Sampling depths for further analyses were set based on the PIA profiles. The PIA was equipped with a syringe rosette sampler (12x60 ml) for targeted depth sampling at 25 cm resolution, integrating about 5 cm water column, allowing for improved resolution compared to conventional Niskin bottle sampling, which integrates over >50 cm. Per depth six syringes were triggered to retrieve enough volume for different analyses.”

1. 308 A volume of 110 mL of water may sound a little bit low to cover the entire microbial communities inhabiting the lake. Could the authors explain this choice of 110 mL?

Answer: Rotsee is a eutrophic lake with cell numbers of $2-6 \times 10^6$ per ml in the mixed layer and an increasing trend within the hypolimnion during our study period, which is relatively high (roughly an order of magnitude higher compared to oligotrophic lakes or most marine pelagic environments). Further, amplicon sequencing requires much less input material than metagenomic or metatranscriptomic sequencing. Also, our PCR worked very well with the input amount of DNA and cDNA from the filtered volume, which is why we considered 110ml enough for our analysis. Further, in this study we did not focus on rare bacteria, but on the abundant members of the bacterial community. For these reasons we think the amount of sample used for our analyses was sufficient. Also, the consistency of results in adjacent layers (Figure 2A, B) provides evidence that the sample size was representative.

1. 321 Have the PCR1 triplicate been sequenced separately? As I understand it, the authors performed only technical replicates (“the first PCR was performed in triplicate”, l. 321) but no sampling replicates (replicates for the same depth of the lake at the same date, sequenced separately). It might have been interesting to perform this kind of sampling replicates to know if the observed microbial communities effectively reflect a true microbial communities in the lake. That being said, the congruence between the different sampling depth is, to a certain extent, convincing.

Answer: Yes we did technical PCR triplicates and pooled them afterwards, which is the recommended strategy for amplicon sequencing. The samples from the mixed layer are very close to replicates because they stem from the same mixed water column and the results agree with this view, which is also why we show min/max and median of the mixed layer samples in Figure 3.

Could the authors explain why they chose to not performed sampling replicates and to not collect water in another place in the same lake to have a better overall picture of what is happening in the whole lake?

Answer: Horizontal mixing in lakes is strong (Lerman and Chou, 1995, page 86), especially in stratified lakes; therefore single location vertical profiles are standard for smaller systems and we think that our profiles are representative for our lake. Taking one profile usually close to the deepest point is common practice in lake surveys (Oswald et al., 2017) and horizontal variation within a lake (excluding near-

shore water or parts with limited water exchange) is expected to be small (Yannarell and Triplett, 2004) in comparison with the vertical variation during stratification and temporal variation of the mixed layer.

1. 376 How did the authors deal with autofluorescence of suspended particles (for CARD-FISH and flow cytometry experiments)?

Answer: Yes, in some aquatic habitats autofluorescence of suspended particles is a common problem for microscopy or flow cytometry (e.g. during high water in a river or floodplain, or in sediment samples), even more so in soil samples. However, in our case we did not encounter this problem. For flow cytometry, the well-established gating strategy for microbial cell counts is designed so that it excludes autofluorescent inorganic particles that have a different signature than stained microbial cells.

FIGURES

Figure 3A Did the maximum, minimum and median relative abundance represent the different values obtained from the technical triplicates? If it is, I am not sure this statistics are really appropriate in this kind of graph.

Answer: The maximum, minimum and median in Figure 3 were not derived from technical triplicates, but are separately measured samples from different depths of the mixed layer.

To clarify what is shown in Figure 3 we added a sentence to the figure caption: "The minimum, maximum and median were derived from separately measured samples from different depths of the mixed layer."

MINOR COMMENTS

ABSTRACT

1. 22 It could be interesting to add a summary of the method used in this study to allow the reader to know what the authors did (16S rRNA gene and pmoA mRNA amplicon sequencing, qPCR, CARD-FISH, potential methane oxidation rate...)

Answer: We added a sentence highlighting our methodological approach.

L22-24: "We investigated this by combining 16S rRNA gene and pmoA mRNA amplicon sequencing, qPCR, CARD-FISH and potential methane-oxidation rates."

RESULTS

1. 81 Since this article focuses on the methane consumption along the depth of the lake, I suggest that the supplementary figure 2 be part of the main figures.

Answer: Thanks for the suggestion. We agree, and we added the methane profiles to the other profiles shown in Figure 2 (new panel Figure 2E).

1. 97-106 I agree with the interest to keep the Figure 2 as the main figure compared with the Supplementary Figure 5 to better see the variations of MOB proportion depending on depth and sampling date. However, since the data for the mixed layer are unreadable in the Figure 2, I suggest to also refer in this section to the Supplementary Figure 5 which represents the proportion of the different MOB lineages among all identified MOB.

Answer: Thanks, we added the reference to Supplementary Fig. 4, in which the changes of the MOB proportions are easier to see.

1. 145 Please add the taxonomic affiliation for ASV_15 as mentioned later in the text (Methylobacter tundipaludum, l. 191) *Answer: We added "Methylobacter" in the text (Line 143).*

1. 148-150 Please refer to Figure 2B and 2C. *Answer: We added the references to the text.*

1. 150 Please refer to Figure 2D *Answer: We added the reference to the text.*

1. 152 Please change (Fig. 2C) to (Fig 2D) *Answer: We changed the reference accordingly.*

DISCUSSION

1. 218 Please change (Zimmermann et al 2019) to the reference number 14.

Answer: We changed the reference accordingly.

METHODS

1. 280 Please add some characteristics of the lake : maximum or average depth, length, width... This allows easier comparison with other lakes

Answer: We added more lake characteristics to the methods. Line 289-290: "Eutrophic lake Rotsee is situated at 419 m a.s.l., has a maximum depth of 16 m, is 2.5 km long and about 200 m wide."

1.300 Did the samples for methane measurements filled with N₂ gas and NaOH pellets directly on the field or did the vials be prepared before going to the field?

Answer: The vials were prepared in the laboratory before going to the field. We added a sentence to the methods to clarify this. Line 311-312: "Samples for methane measurements were injected into airtight 40 ml serum vials filled with N₂ gas and 4 g NaOH pellets (≥98% purity, Sigma-Aldrich, Darmstadt, Germany) for preservation. The vials were prepared in the laboratory prior to field work."

1. 327 Have the libraries been pooled equimolarly?

Answer: Yes. We added this information. Line 345-347: "Libraries were purified with Agencourt AMPure XP kit (Beckman coulter, Indianapolis, IN, USA), quantified (Qubit, Thermo Fischer Scientific) and pooled equimolarly."

1. 333 How many sequences were used to build the ASV table? Have the number of sequences been normalized at a given number of sequences per sample?

Answer: All samples used in analysis had at least 5000 reads. We did normalize the sample sums to 1 to obtain relative abundances. We clarified this in the methods section. In our study we did not focus on rare species or alpha diversity, which are sensitive to sequencing depth therefore we did not rarefy our sequencing data.

Line 350-358: "Amplicon sequence variants (ASVs) were inferred with the DADA2 pipeline³⁵ (v1.6.0) in R36 (v3.4.1). 16S rRNA reads were trimmed to 270 nt (forward) and 210 nt (reverse) and pmoA reverse reads were trimmed to 235 nt. At the first instance of a quality score of less or equal to two the read was truncated and reads with ambiguous bases or an expected error rate $EE = \sum(10^{-(Q/10)})$ exceeding three were removed. After calculating error rates the reads were dereplicated and denoised to infer exact sequence variants. After merging forward and reverse reads chimera were removed. Samples with less than 5000 reads were removed. Taxonomy of 16S rRNA ASVs was assigned using the SILVA reference database (v132)³⁷ and reads assigned to mitochondria or chloroplasts were removed with phyloseq³⁸ (1.24.2) in R (3.5.0) and the read counts were transformed to relative abundances."

1. 343 Which type of alignment has been used to build the phylogenetic trees?

Answer: We aligned the sequences with Muscle in MEGA7. This information has been added to the methods section.

Line 365-368: "Alignment (Muscle) and neighbor-joining trees were constructed in MEGA7 with 10000 bootstrap replications using Jukes-Cantor evolutionary distance for 16S rRNA sequences or Poisson correction method for pmoA amino acid sequences."

1. 376 Which fluorophores have been used of the CARD-FISH experiments?

Answer: We used Oregon Green 488 labelled tyramide. We clarified this in the methods part.

Line 393-400: "The horseradish peroxidase labelled probes Mg84, Mg669 and Mg705⁴¹ were mixed to target type I MOB. As control probes EUB338I-III (universal bacterial probe, positive control) and NON338 (negative control)^{42,43} were separately applied to each sample. Filters were incubated at 46°C for 2.5 h with 150 µl of the respective hybridization buffer (20% formamide Mg-mix and 35% for EUB and NON) containing 1 µl of probe (50 ng µl⁻¹). After washing, filters were placed into the amplification buffer containing tyramide labelled with Oregon Green 488 at 37°C for 30 min in the dark. DAPI was used for counterstaining and the filters were mounted on glass slides in a mix of Citifluor and Vectashield (4:1)."

1. 384 Maybe the authors should add some CARD-FISH pictures in supplementary data to show which type of marked cells was counted with Daime.

Answer: We added an exemplary Figure of how the cells were identified with daime. (Supplementary Figure 5). We also took the opportunity to provide more detail on the cell counting procedure in the legend to this figure.

FIGURES

Figure 2 I suggest to add affiliations in the caption. For example, ASV_5 / CAB2E06 and ASV_354 / Methylocystis... This would greatly facilitate reading.

Answer: Thanks for this suggestion; we added the ASV affiliations to the figure legend (Figure 2 and 3) to facilitate reading.

Figure 3A As I understand, the Figure 3A represent another way to visualize the data presenting in the Figure 2A.

Answer: Yes, we added this figure to visualize the mixed layer dynamics of MOB assemblage in comparison to methane oxidation rates and physicochemical parameters.

For greater clarity, the authors could only put on this graph the main MOB lineages (ASV_5, ASV_12, ASV_15, ASV_8, ASV_4). The other MOB lineages are unreadable on this graph and it is thus difficult to read.

Answer: Thanks, we absolutely agree with this suggestion; we now focus on the 5 most common MOB lineages of our data set, which improved the clarity of the figure.

Reviewer #2 (Remarks to the Author):

The manuscript by Mayr et al. describes the changes in the community composition and CH₄ oxidation capacity of the methanotrophic community inhabiting lake Rotsee during the autumn overturn period. The manuscript is well written and analyses appropriate. However, the authors are overselling a bit the story and it's novelty, and downgrading the value of earlier studies. Thus, some of the statements about the novelty of the study should be toned down.

Answer: Thank you for your valuable comments and suggestions.

We toned down some statements within Abstract, Introduction and Discussion section in order to avoid overselling of the study (see detailed comments below).

To the discussion should be added how realistic the measured CH₄ oxidation rates are compared to the conditions in the nature.

Answer: Yes this is an interesting aspect. The measured methane oxidation rates are potential rates, equivalent to maximum methane oxidation rates to CO₂, because the natural MOB assemblage was provided with non-limiting O₂ and methane concentrations. They are introduced as such in the manuscript (Methods section: Line 373, 377). In the context of our study these potential rates are of great interest, because they show how much methane could be oxidized as soon as methane is introduced into the mixed layer. These potential rates will often not be realized in the lake, thus the MOB assemblage is acting below its potential. In a parallel study (Zimmermann et al. submitted) we analysed this aspect in more detail, where we found that according to the growth model MOB grew always below their maximum growth rate.

We opted to use potential methane oxidation capacity, to acknowledge that the capacity is based on potential rates, and we added a clarifying sentence Line 196-198 that refers to the parallel paper: "Detailed analysis revealed that potential methane oxidation capacity always exceeded the methane input from the hypolimnion and limited the emitted methane to a small percentage of the stored methane during lake overturn¹⁴"

This is in addition to the discussion of the methane oxidation capacity (which is based on the potential methane oxidation rates) which was already present in the discussion, and which we amended as follows:

Line 248-254: “While potential methane oxidation rates may overestimate in-situ rates under limiting methane concentrations, our oxidation capacity estimate is conservative as only potential methane oxidation to CO₂ was measured and incorporation into biomass is not included. Even though the MOB growth rates seem low compared to observations in MOB isolates^{21,22} with doubling times of several hours, they may be in line with the environmental conditions: temperatures in the mixed layer dropped from 18.4 (Oct 4) to 6.1°C (Dec 12), thus slowing process and growth rates are expected. Further, persistently low methane concentrations likely reduced effective growth rates.”

MINOR COMMENTS

L25-27 I would not say that this is a “previously unknown” phenomenon. For example, Taipale et al (AME 2011), which is cited elsewhere in the ms, already stated that methanotroph abundance was especially high during autumn overturn.

Answer: Yes, there is some previous knowledge on this topic in the literature, and we now acknowledge Taipale et al. in Line 53 “Taipale et al. noted an increase of methane oxidation rate and abundance of type I methanotrophs linked to the autumn lake mixing period⁵, but the dynamics of different MOB taxa have not been studied.” Our study does, however, add several new aspects. For example previous studies could not show that the MOB cells and the increasing methane oxidation capacity is built up by a successional changing MOB assemblage, which allowed us to conclude that newly grown MOB cells, rather than persisting cells or cells from the hypolimnion build the high methane oxidation capacity which we observed. Further, in a parallel study we show the methane emission from the same period and modelled the methane budget, which is why we can link the succession and bloom of the MOB assemblage to the amount of methane outgassing from the lake.

Nevertheless, we removed the “previously unknown” in order not to oversell the story.

L37-39 Could you please provide some citation to this? I’m only aware of literature stating that carbon transfer from CH₄ to biomass in oxic conditions is an important energy source for the lake food web (e.g. Jones & Grey 2011, Freshwater Biology) and reports of relatively high C uptake to biomass (e.g. He et al. 2015, Environmental Microbiology).

Answer: Thanks for pointing out this mistake. The carbon conversion efficiency of methane incorporation into biomass is actually rather variable in the literature (19%-70%, Leak & Dalton 1986, Appl Microbiol Biotechnol). What we wanted to say is however only that both CO₂ and biomass are produced. We corrected this in the text.

Line 38- 40: “According to the second hypothesis, methanotrophs oxidize most of the methane with oxygen, thereby forming biomass and CO₂, a GHG which has a 34 times lower global warming potential for a 100-yr time-scale⁴.

L274-277 Since the amount of oxidized CH₄ is in the same range as in previous studies (L256), “extraordinarily” is quite strong term. Also, it suggests that lake here would not be representative, which would mean that no generalization should be done based on this study. In my opinion “high CH₄ oxidation capacity” would be strong enough statement.

Answer: Thanks we removed “extraordinarily” as suggested.

L331 You mean the reads were cut at the position where the quality score dropped below 2?

Answer: Yes, we clarified the DADA2 analysis methods section accordingly:

Line 347-349: At the first instance of a quality score of less or equal to two the read was truncated and reads with ambiguous bases or an expected error rate $EE = \sum(10^{-(Q/10)})$ exceeding three were removed.

L332 Exceeding three what? %? Bases?

Answer: 3 bases. Line 352-345: At the first instance of a quality score of less or equal to two the read was truncated and reads with ambiguous bases or an expected error rate $EE = \sum(10^{-(Q/10)})$ exceeding three were removed.

L332 Were the forward and reverse reads assembled or analysed separate?

Answer: Forward and reverse reads were assembled. We clarified this in the methods section.

Line 354-355: "After calculating error rates the reads were dereplicated and denoised to infer exact sequence variants. After merging forward and reverse reads chimera were removed."

L350-358 This part would make more sense to be before 16S and pmoA analysis section, since the data processing from this material is explained there.

Answer: To clarify: In the "pmoA gene and transcript quantification" section we explain the methods of the quantitative PCR, which is not connected to the sequencing for which a separate amplification is performed. We nevertheless moved it before the amplicon analysis in the methods section, because it makes more sense in terms of the increasing complexity of analysis.

In Line 92 (Results section) we have changed "pmoA mRNA sequencing combined with qPCR..." to "The sequencing of pmoA mRNA and the quantification of pmoA transcripts with qPCR..." to avoid confusion.

L364 Please add how close to in situ temperature the incubations were done and to which direction the deviation was.

Answer: We added the incubation temperatures to the methods section. Further, we clarified how the incubation temperature was chosen. Line 379-382: "The incubation temperature for each date was chosen based on the in-situ temperature profile at the date of sampling. For the first three dates a temperature between the mixed layer and lake bottom temperature (~6.5°C) was chosen, for later dates the mixed layer temperature was chosen (incubation temperatures from Oct 4th to Jan 23rd in °C: 13, 11, 11, 11, 8.9, 7, 6, 4)."

Reviewer #3 (Remarks to the Author):

In the manuscript "Growth and rapid succession of methanotrophs effectively limit methane release during lake overturn" the authors report their findings on the change in community structure and abundance of methane-oxidizing bacteria (MOB) as well as potential methane oxidation activity along the depth profile of Lake Rotsee during the annual lake overturn.

As mixing proceeds, the community of MOB is changing and minor players during the stratified period outgrow the other MOB. The increase in MOB abundance results in a significantly increased potential methane oxidation capacity in the lake and limits methane release into the atmosphere. This study nicely combines complementary methods to support the findings. 16S rRNA gene-based methods are complemented by pmoA mRNA-based quantification and CARD-FISH and backed up with potential methane oxidation rate measurements and nutrient data. Together, these data combine to a compelling picture of the dynamics in MOB community during a gradual mixing event. Although the study was limited to one lake, it clearly shows that methane stored in the anoxic hypolimnion of lakes is not released completely to the atmosphere upon mixing. In addition, the study emphasizes that MOB communities can be very dynamic and that different MOB clearly occupy different niches! These findings are clearly interesting to the community.

I only have a few specific comments:

1) In the color printout I used to review the paper, the different shades of gray in Fig 1B and C are very hard to distinguish, especially the darker shades. I suggest using additional color to increase the contrast between the different lines - especially since many figures already use color.

Answer: Thanks - we tried to improve the clarity of Figure 1B and 1C as much as possible and added a color gradient, as suggested, to increase the contrast between the different lines.

2) The grey shading in Fig. 3C makes the numbers really hard to read in my printout. I suggest changing the font to bold to increase the contrast or skip the shading. In my opinion, the shading does not add additional information as the table is short enough to actually read the numbers.

Answer: Thanks – to improve readability we changed numbers with light background to black, but keep the shading because we think it allows readers to quickly grasp the changes of the different parameters.

3) The sampling year in the main text and the methods is contradicting. Sometimes it's 2016, sometimes 2017.

Answer: Yes, thanks for pointing this out. We sampled from October 2016 to January 2017. We checked and corrected the sampling year accordingly throughout the manuscript.

4) Just a suggestion: Would it be possible to calculate a maximum growth rate of the MOB community using growth yield data from the literature and methane flux data you can calculate from your measurements? I would be interested to see whether the calculated max growth rates based on the amount of methane mixed into the mixed layer roughly match the observed rates. Would be a nice addition if they do, but would only distract from the main message of the paper if they do not.

Answer: Yes, indeed these are interesting questions, which we investigated in detail in a parallel study in which we looked at the mass balances between methane mixed into the mixed layer, methane oxidation rates, methane emission and biomass build-up and the robustness of the methane oxidation during lake overturn: "Lake overturn as a key driver for methane oxidation" Zimmermann et al. (submitted). Further, we have a study underway looking at the methane affinity of the MOB assemblage during lake overturn, in which we will also answer some of these questions.

We agree that these aspects are interesting, but also require extensive analysis/modelling and additional data, which we will present in two future studies. The Zimmermann study is referenced in the manuscript in Line 196-198.

5) I was not aware of methane released by dissolving NaOH pellets. Thanks for adding this information! *Answer: Yes, this was unexpected for us.*

6) The flow cytometry data mentioned in the methods are not in the main article, but seem to fit well together with the type1 MOB CARD-FISH and 16S gene copy number data. Is there a reason not to report relative CARD-FISH/particles numbers as you did with the MOB-general 16S ratio?

Answer: Yes, it would be possible to calculate a ratio of CARD-FISH MOB to flow cytometry bacterial counts, but a similar measure - the relative abundance of MOB of 16S rRNA sequences - is already shown. The flow cytometry cell numbers were used to inform us of how much volume to filter on the microscopy filters for CARD-FISH and they are there for the readers to put the 16S rRNA gene relative abundances into context (Flow cytometry counts are shown in the table of Figure 3C). We actually did the CARD-FISH analysis precisely so we can directly show growth of MOB, the proof of which requires absolute cell abundances, whereas relative abundances, which are generally produced by 16S rRNA gene sequencing cannot conclusively do this.

We added a sentence highlighting the microbial cell numbers in the results:

L180-181: "The microbial cell numbers in the mixed layer were lower in December (Fig. 3C), but at the same time MOB percentage and absolute MOB cell numbers peaked (Fig. 3A, Fig. 2D)."

REVIEWERS' COMMENTS:

Reviewer #1 (Remarks to the Author):

I thank the authors for their detailed answers on the different concerns I had. They have done a great job with the revised version of the manuscript and have addressed (or defended) all of my comments reasonably. Looking forward to reading the study online!

Reviewer #2 (Remarks to the Author):

The authors have addressed all comments carefully and thoroughly. Only one minor comment:

Figure 1. The last sentence of legend, should be colors in B and C, not A and B.

Reviewer #3 (Remarks to the Author):

Thank you very much for incorporating the changes suggested by all reviewers. I think the manuscript improved, and I am completely satisfied with the changes. Only one tiny remark:

The abbreviation "mio" for million is not very common. Maybe write 10^6 instead (Fig 3).

Please find below our replies to the remaining minor comments of the reviewers.

Reviewer #1 (Remarks to the Author):

I thank the authors for their detailed answers on the different concerns I had. They have done a great job with the revised version of the manuscript and have addressed (or defended) all of my comments reasonably. Looking forward to reading the study online!

- We thank for the valuable input

Reviewer #2 (Remarks to the Author):

The authors have addressed all comments carefully and thoroughly. Only one minor comment:

Figure 1. The last sentence of legend, should be colors in B and C, not A and B.

Thank you, corrected in legend to Figure 1. Note that all panel labels were changed to lowercase letters to conform to journal standards.

Reviewer #3 (Remarks to the Author):

Thank you very much for incorporating the changes suggested by all reviewers. I think the manuscript improved, and I am completely satisfied with the changes. Only one tiny remark:

The abbreviation "mio" for million is not very common. Maybe write 10^6 instead (Fig 3).

Thank you, we have corrected this as suggested